# PI3NN: Out-of-distribution-aware prediction intervals from three neural networks

**Siyan Liu, Pei Zhang & Dan Lu**
Computational Sciences and Engineering Division
Oak Ridge National Laboratory
1 Bethel Valley Rock, Oak Ridge, TN 37830, USA
`{lius1, zhangp1, lud1}@ornl.gov`

**Guannan Zhang**[*]
Computer Science and Mathematics Division
Oak Ridge National Laboratory
1 Bethel Valley Rock, Oak Ridge, TN 37830, USA
`{zhangg}@ornl.gov`

## Abstract

We propose a novel prediction interval (PI) method for uncertainty quantification, which addresses three major issues with the state-of-the-art PI methods. First, existing PI methods require retraining of neural networks (NNs) for every given confidence level and suffer from the crossing issue in calculating multiple PIs. Second, they usually rely on customized loss functions with extra sensitive hyperparameters for which fine tuning is required to achieve a well-calibrated PI. Third, they usually underestimate uncertainties of out-of-distribution (OOD) samples leading to over-confident PIs. Our PI3NN method calculates PIs from linear combinations of three NNs, each of which is independently trained using the standard mean squared error loss. The coefficients of the linear combinations are computed using root-finding algorithms to ensure tight PIs for a given confidence level. We theoretically prove that PI3NN can calculate PIs for a series of confidence levels without retraining NNs and it completely avoids the crossing issue. Additionally, PI3NN does not introduce any unusual hyperparameters resulting in a stable performance. Furthermore, we address OOD identification challenge by introducing an initialization scheme which provides reasonably larger PIs of the OOD samples than those of the in-distribution samples. Benchmark and real-world experiments show that our method outperforms several state-of-the-art approaches with respect to predictive uncertainty quality, robustness, and OOD samples identification.

## 1 Introduction

Neural networks (NNs) are widely used in prediction tasks due to their unrivaled performance in modeling complex functions. Although NNs provide accurate predictions, quantifying the uncertainty of their predictions is a challenge. Uncertainty quantification (UQ) is crucial for many real-world applications such as self-driving cars and autonomous experimental and operational controls. Moreover, effectively quantified uncertainties are useful for interpreting confidence, capturing out-of-distribution (OOD) data, and realizing when the model is likely to fail.

A diverse set of UQ approaches have been developed for NNs, ranging from fully Bayesian NNs [1], to assumption-based variational inference [2; 3], and to empirical ensemble approaches [4; 5; 6]. These methods require either high computational demands or strong assumptions or large memory costs. Another set of UQ methods is to calculate prediction intervals (PIs), which provides a lower and upper bound for an NN's output such that the value of the prediction falls between the bounds for some target confidence level $\gamma$ (e.g., 95%) of the unseen data. The most common techniques to construct the PI are the delta method (also known as analytical method) [7; 8], methods that directly predict the variance (e.g., maximum likelihood methods and ensemble methods) [9; 10] and quantile regression methods [11; 12]. Most recent PI methods are developed on the high-quality principle—a PI should be as narrow as possible, whilst capturing a specified portion of data. Khosravi et al. [13] developed the Lower Upper Bound Estimation method by incorporating the high-quality principle directly into the NN loss function for the first time. Inspired by [13], the Quality-Driven (QD) prediction interval approach in [14] defines a loss function that can generate a high-quality PI

---

[*]Corresponding author.

and is able to optimize the loss using stochastic gradient descent as well. Built on QD, the Prediction Intervals with specific Value prEdictioN (PIVEN) method in [15] adds an extra term in the loss to enable the calculation of point estimates and the PI method in [16] further integrates a penalty function to the loss to improve the training stability of QD. The Simultaneous Quantile Regression (SQR) method [17] proposes a loss function to learn all quantiles of a target variable with one NN. Existing PI methods suffer from one or more of the following limitations: (1) Requirement of NNs retraining for every given confidence level $\gamma$ and suffering from the crossing issue [18] when calculating PIs for differnt $\gamma$; (2) Requirement of hyper-parameter fine tuning; (3) Lack of OOD identification capability resulting in unreasonably narrow PIs for OOD samples (See Section 2.1 for details).

To address these limitations, we develop PI3NN (prediction interval based on three neural networks)— a novel method for calculating PIs. We first lay out the theoretical foundation of the PI3NN in Section 3.1 by proving Lemma 1 that connects the ground-truth upper and lower bounds of a PI to a family of models that are easy to approximate. Another advantage of the model family introduced in Lemma 1 is that it makes the NN training independent of the target confidence level, which makes it possible to calculate multiple PIs for a series of confidence levels without retraining NNs. On the basis of the theoretical foundation, we describe the main PI3NN algorithm in Section 3.2. Different from existing PI methods [14; 15; 16; 17] that design complicated loss functions to obtain a well-calibrated PI by fine-tuning their loss hyper-parameters, our method simply uses the standard mean squared error (MSE) loss for training. Additionally, we theoretically prove that PI3NN has a non-crossing property in Section 3.2.1. Moreover, we address the OOD identification challenge by proposing a simple yet effective initialization scheme in Section 3.3, which provides larger PIs of the OOD samples than those of the in-distribution (InD) samples.

The main contributions of this work are summarized as follows:

1. Our PI3NN method can calculate PIs for a series of confidence levels without retraining NNs; and the calculated PIs completely avoid the crossing issue as proved in Section 3.2.1.

2. The theoretical foundation in Section 3.1 enables PI3NN to use the standard MSE loss to train three NNs without introducing extra hyper-parameters that need to be fine-tuned for a good PI.

3. We develop a simple yet effective initialization scheme and a confidence score in Section 3.3 to identify OOD samples and reasonably quantify their uncertainty.

## 1.1 RELATED WORK

*Non-PI approaches for UQ.* Early and recent work was nicely summarized and reviewed in these three survey papers [19; 20; 21]. The non-PI approaches use a distribution to quantify uncertainty, which can be further divided into Bayesian [1] and non-Bayesian methods. Bayesian methods—including Markov chain Monte Carlo [22] and Hamiltonian Monte Carlo [23]—place priors on NN weights and then infer a posterior distribution from the training data. Non-Bayesian methods includes evidential regression [6] that places priors directly over the likelihood function and some ensemble learning methods that do not use priors. For example, the DER method proposed in [6] placed evidential priors over the Gaussian likelihood function and training the NN to infer the hyperparameters of the evidential distribution. Gal and Ghahramani [3] proposed using Monte Carlo dropout to estimate predictive uncertainty by using Dropout (which can be interpreted as ensemble model combination) at test time. Deep ensembles [4] employed a combination of ensembles of NNs learning and adversarial training to quantify uncertainty with a Gaussian distributional assumption on the data. Pearce et al. [5] proposed an anchored ensembling by using the randomized MAP sampling to increase the diversity of NN training in the ensemble.

## 2 BACKGROUND

We are interested in building PIs for the output of the regression problem $y = f(\boldsymbol{x}) + \varepsilon$ from a training set $\mathcal{D}_{\text{train}} = \{(\boldsymbol{x}_i, y_i)\}_{i=1}^{N}$, where $\boldsymbol{x} \in \mathbb{R}^d$, $y \in \mathbb{R}$, and $\varepsilon$ is the random noise. We do not impose distributional assumption on the noise $\varepsilon$. Since the output of $f(\boldsymbol{x})$ is polluted by the noise, the output $y$ of the regression model $y = f(\boldsymbol{x}) + \varepsilon$ is also a random variable. For a given confidence level $\gamma \in (0, 1)$, the ground-truth $100\gamma\%$ PI, denoted by $[L_\gamma^{\text{true}}(\boldsymbol{x}), U_\gamma^{\text{true}}(\boldsymbol{x})]$, is defined by

$$\mathbb{P}[L_\gamma^{\text{true}}(\boldsymbol{x}) \leq y \leq U_\gamma^{\text{true}}(\boldsymbol{x})] = \gamma. \tag{1}$$

Note that $L_\gamma^{\text{true}}(\boldsymbol{x})$ and $U_\gamma^{\text{true}}(\boldsymbol{x})$ are not unique for a fixed $\gamma$ in the definition of Eq. (1), because the probability of $y$ outside the PI, i.e., $\mathbb{P}[y > U_\gamma^{\text{true}}(\boldsymbol{x})$ or $y < L_\gamma^{\text{true}}(\boldsymbol{x})]$, may be split in any way between the two tails. In this work, we aim to approximate the PI that satisfies

$$\mathbb{P}[y > U_\gamma^{\text{true}}(\boldsymbol{x})] = (1-\gamma)/2 \quad \text{and} \quad \mathbb{P}[y < L_\gamma^{\text{true}}(\boldsymbol{x})] = (1-\gamma)/2, \tag{2}$$

which is *unique* because the probability outside the PI is equally split between the two tails.

## 2.1 MOTIVATION

Recent effort on PI methods, e.g., QD, SQR and PIVEN, tend to exploit the NNs to learn the upper and lower bounds in Eq. (2). Despite their promising performance, these methods suffer from some unsatisfactory drawbacks. This effort is motivated by their following limitations:

- *Requirement of retraining NNs for every given $\gamma$ and the crossing issue when calculating multiple PIs.* Existing PI methods usually incorporate $\gamma$ into their loss functions for training NNs, so that each NN can only predict PI for a specific $\gamma$, which is not convenient for users. On the other hand, even multiple NNs can be trained for PIs with multiple $\gamma$ values, the approximate PIs often encounter the crossing issue, e.g., the upper bounds for different $\gamma$ values may cross each other, which is not reasonable. To alleviate this issue, a non-crossing constraint is usually added to the loss as a regularization to encourage non-crossing PIs. However, the non-crossing constraint may deteriorate the quality of the approximate PI, because due to the trade-off between the original loss and the non-crossing constraint.

- *Requiring hyper-parameter fine tuning.* Recently developed PI methods [14; 15; 16] tend to design complicated loss functions to obtain a well-calibrated PI. Although these work has achieved promising results, their performance is sensitive to the unusual hyperparameters introduced into their customized loss functions. Thus, hyperparameter fine tuning is usually required for each specific problem to achieve satisfactory upper and lower bounds.

- *Lack of OOD identification capability.* OOD identification is a critical metric to evaluate the performance of an UQ method. It has been received significant attention in recent UQ method development in the machine learning community. However, the OOD identification has not been deeply studied for PI methods in solving the regression problem. Even though there are some promising empirical results on OOD-aware PIs [15], the underlying mechanism is still not clear, making it difficult to extend the occasional success to a general setting.

## 3 THE PI3NN METHOD

The main contribution is presented in this section. Section 3.1 shows a theoretical justification of our method, where Lemma 1 plays a critical role to connect the ground-truth upper and lower bounds to Eq. (5) and Eq. (6) that are easier to approximate. Section 3.2 introduces the main PI3NN algorithm inspired by Lemma 1. Section 3.3 describes how to turn on the OOD identification capability in the PI3NN method. The features of our methods are illustrated using a simple example in Section 3.4.

## 3.1 THEORETICAL JUSTIFICATION

To proceed, we first rewrite Eq. (2) to an equivalent form

$$\mathbb{E}\big[\mathbf{1}_{y>U_\gamma^{\text{true}}(\boldsymbol{x})}\big] - (1-\gamma)/2 = 0 \quad \text{and} \quad \mathbb{E}\big[\mathbf{1}_{y<L_\gamma^{\text{true}}(\boldsymbol{x})}\big] - (1-\gamma)/2 = 0, \tag{3}$$

where $\mathbf{1}_{(.)}$ is the indicator function, defined by

$$\mathbf{1}_{y>U_\gamma^{\text{true}}(\boldsymbol{x})} = \begin{cases} 1, & \text{if } y > U_\gamma^{\text{true}}(\boldsymbol{x}), \\ 0, & \text{otherwise,} \end{cases} \quad \text{and} \quad \mathbf{1}_{y<L_\gamma^{\text{true}}(\boldsymbol{x})} = \begin{cases} 1, & \text{if } y < L_\gamma^{\text{true}}(\boldsymbol{x}), \\ 0, & \text{otherwise.} \end{cases} \tag{4}$$

For simplicity, we take $U_\gamma^{\text{true}}(\boldsymbol{x})$ as an example in the rest of Section 3.1, and the same derivation can be applied to $L_\gamma^{\text{true}}(\boldsymbol{x})$. In definition, $U_\gamma^{\text{true}}(\boldsymbol{x})$ has the following three properties:

(P1) $U_\gamma^{\text{true}}(\boldsymbol{x}) \geq \mathbb{M}[y] = f(\boldsymbol{x}) + \mathbb{M}[\varepsilon]$, where $\mathbb{M}[\cdot]$ denotes the median of a random variable,

(P2) $U_\gamma^{\text{true}}(\boldsymbol{x}) - f(\boldsymbol{x})$ is independent of $\boldsymbol{x}$, because $\varepsilon$ is independent of $\boldsymbol{x}$,

(P3) $U_\gamma^{\text{true}}(\boldsymbol{x})$ is unique for a given confidence level $\gamma \in (0, 1)$.

Next, we show that the ground-truth $U_\gamma^{\text{true}}(\boldsymbol{x})$ is a member of the following model family:

$$\hat{U}(\boldsymbol{x}|\alpha) = \mathbb{M}[y] + \alpha \, \mathbb{E}\left[(y - \mathbb{M}[y])\mathbf{1}_{y - \mathbb{M}[y] > 0}\right], \tag{5}$$

where $\alpha \geq 0$ is a scalar model parameter and $\mathbb{M}[\cdot]$ denotes the median of $y$. To this end, we prove the following lemma:

**Lemma 1.** *For a given $\gamma \in (0, 1)$, there exists a unique $\alpha(\gamma)$, such that $\hat{U}(\boldsymbol{x}|\alpha(\gamma)) = U_\gamma^{\text{true}}(\boldsymbol{x})$.*

The proof of this lemma is given in Appendix A. The same result can be derived for the lower bound, i.e., for a given $\gamma \in (0, 1)$, there exists a unique $\beta(\gamma)$, such that $\hat{L}(\boldsymbol{x}|\beta(\gamma)) = L_\gamma^{\text{true}}(\boldsymbol{x})$, where $\hat{L}(\boldsymbol{x}|\beta)$ is defined by

$$\hat{L}(\boldsymbol{x}|\beta) = \mathbb{M}[y] - \beta \, \mathbb{E}\left[(\mathbb{M}[y] - y)\mathbf{1}_{\mathbb{M}[y] - y > 0}\right]. \tag{6}$$

Lemma 1 connects the ground-truth upper and lower bounds to the model families $\hat{U}(\boldsymbol{x}|\alpha)$ and $\hat{L}(\boldsymbol{x}|\beta)$, such that the task of approximating $U_\gamma^{\text{true}}(\boldsymbol{x})$ and $L_\gamma^{\text{true}}(\boldsymbol{x})$ can be decomposed into two sub-tasks:

- Approximate the models $\hat{U}(\boldsymbol{x}|\alpha)$ and $\hat{L}(\boldsymbol{x}|\beta)$ by training NNs;
- Calculate the optimal values of $\alpha(\gamma)$ and $\beta(\gamma)$ for any $\gamma \in (0, 1)$.

In above theoretical justification we assume the noise $\varepsilon$ is homoscedastic as commonly done in regression problems. But note that we use this assumption just for simplifying the theoretical proof. In practice, our method can be generally applied to problems with different forms of noise.

## 3.2 THE MAIN ALGORITHM

PI3NN accomplishes the above two sub-tasks in four steps. *Step 1-3* build approximations of $\hat{U}(\boldsymbol{x}|\alpha)$ and $\hat{L}(\boldsymbol{x}|\beta)$ by training three independent NNs, denoted by $f_{\boldsymbol{\omega}}(\boldsymbol{x})$, $u_{\boldsymbol{\theta}}(\boldsymbol{x})$, $l_{\boldsymbol{\xi}}(\boldsymbol{x})$, which approximate $\mathbb{M}[y]$, $\mathbb{E}[(y - \mathbb{M}[y])\mathbf{1}_{y - \mathbb{M}[y] > 0}]$, and $\mathbb{E}[(\mathbb{M}[y] - y)\mathbf{1}_{\mathbb{M}[y] - y > 0}]$, respectively. After the three NNs are trained, we calculate the optimal values of $\alpha(\gamma)$ and $\beta(\gamma)$ in *Step 4* using root-finding techniques.

*Step 1: Train $f_{\boldsymbol{\omega}}(\boldsymbol{x})$ to approximate the mean $\mathbb{E}[y]$.* This step follows the standard NN-based regression process using the standard MSE loss. The trained $f_{\boldsymbol{\omega}}(\boldsymbol{x})$ will serve two purposes. The first is to provide a baseline to approximate $\mathbb{M}[y]$ in *Step 2*; the second is to provide a point estimate of $\mathbb{E}[f]$. In this step, we use the standard $L_1$ and $L_2$ regularization to avoid over-fitting.

*Step 2: Add a shift $\nu$ to $f_{\boldsymbol{\omega}}(\boldsymbol{x})$ to approximate the median $\mathbb{M}[y]$.* A scalar $\nu$ is added to $f_{\boldsymbol{\omega}}(\boldsymbol{x})$, such that each of the two data sets $\mathcal{D}_{\text{upper}}$ and $\mathcal{D}_{\text{lower}}$, defined by

$$\begin{aligned} \mathcal{D}_{\text{upper}} &= \left\{(\boldsymbol{x}_i, y_i - f_{\boldsymbol{\omega}}(\boldsymbol{x}_i) - \nu) \mid y_i \geq f_{\boldsymbol{\omega}}(\boldsymbol{x}_i) + \nu, i = 1, \ldots, N\right\}, \\ \mathcal{D}_{\text{lower}} &= \left\{(\boldsymbol{x}_i, f_{\boldsymbol{\omega}}(\boldsymbol{x}_i) + \nu - y_i) \mid y_i < f_{\boldsymbol{\omega}}(\boldsymbol{x}_i) + \nu, i = 1, \ldots, N\right\}, \end{aligned} \tag{7}$$

contains 50% of the total training samples in $\mathcal{D}_{\text{train}}$. Note that $\mathcal{D}_{\text{upper}}$ and $\mathcal{D}_{\text{lower}}$ include data points above and below $f_{\boldsymbol{\omega}}(\boldsymbol{x}) + \nu$, respectively. The value of the shift $\nu$ is calculated using a root-finding method [24] (e.g., the bisection method) to find the root (i.e., the zero) of the function $Q(\nu) = \sum_{(\boldsymbol{x}_i, y_i) \in \mathcal{D}_{\text{train}}} \mathbf{1}_{y_i > f_{\boldsymbol{\omega}}(\boldsymbol{x}_i) + \nu} - 0.5N$. This is similar to finding the roots of Eq. (10), so we refer to Lemma 2 for the existence of the root.

*Step 3: Train $u_{\boldsymbol{\theta}}(\boldsymbol{x})$ and $l_{\boldsymbol{\xi}}(\boldsymbol{x})$ to learn $\mathbb{E}[(y - \mathbb{M}[y])\mathbf{1}_{y - \mathbb{M}[y] > 0}]$ and $\mathbb{E}[(\mathbb{M}[y] - y)\mathbf{1}_{\mathbb{M}[y] - y > 0}]$.* We use $\mathcal{D}_{\text{upper}}$ to train $u_{\boldsymbol{\theta}}(\boldsymbol{x})$, and use $\mathcal{D}_{\text{lower}}$ to train $l_{\boldsymbol{\xi}}(\boldsymbol{x})$. To ensure the outputs of $u_{\boldsymbol{\theta}}(\boldsymbol{x})$ and $l_{\boldsymbol{\xi}}(\boldsymbol{x})$ are positive, we add the operation $\sqrt{(\cdot)^2}$ to the output layer of both NNs. The two NNs are trained separately using the standard MSE loss, i.e.,

$$\boldsymbol{\theta} = \arg\min_{\boldsymbol{\theta}} \sum_{\mathcal{D}_{\text{upper}}} (y_i - f_{\boldsymbol{\omega}}(\boldsymbol{x}_i) - \nu - u_{\boldsymbol{\theta}}(\boldsymbol{x}_i))^2, \quad \boldsymbol{\xi} = \arg\min_{\boldsymbol{\xi}} \sum_{\mathcal{D}_{\text{lower}}} (f_{\boldsymbol{\omega}}(\boldsymbol{x}_i) + \nu - y_i - l_{\boldsymbol{\xi}}(\boldsymbol{x}_i))^2. \tag{8}$$

*Step 4: Calculate the PI via root-finding methods.* Using the three NNs trained in *Step 1-3*, we build approximations of $\hat{U}(\boldsymbol{x}|\alpha)$ and $\hat{L}(\boldsymbol{x}|\alpha)$, denoted by

$$U(\boldsymbol{x}|\alpha) = f_{\boldsymbol{\omega}}(\boldsymbol{x}) + \nu + \alpha u_{\boldsymbol{\theta}}(\boldsymbol{x}), \qquad L(\boldsymbol{x}|\beta) = f_{\boldsymbol{\omega}}(\boldsymbol{x}) + \nu - \beta l_{\boldsymbol{\xi}}(\boldsymbol{x}). \tag{9}$$

For a sequence of confidence levels $\Gamma = \{\gamma_k\}_{k=1}^K$, we use the bisection method [24] to calculate the optimal value of $\alpha(\gamma_k)$ and $\beta(\gamma_k)$ by finding the roots (i.e., the zeros) of the following functions:

$$
\begin{aligned}
Q_{\text{upper}}(\alpha) &= \sum_{(\boldsymbol{x}_i, y_i) \in \mathcal{D}_{\text{upper}}} \mathbf{1}_{y_i > U(\boldsymbol{x}_i|\alpha)} - \lceil N(1-\gamma_k)/2 \rceil, \\
Q_{\text{lower}}(\beta) &= \sum_{(\boldsymbol{x}_i, y_i) \in \mathcal{D}_{\text{lower}}} \mathbf{1}_{y_i < L(\boldsymbol{x}_i|\beta)} - \lceil N(1-\gamma_k)/2 \rceil,
\end{aligned}
\tag{10}
$$

for $k = 1, \ldots, K$. Note that Eq. (10) can be reviewed as the discrete version of Eq. (3) and/or Eq. (**??**). When $\mathcal{D}_{\text{upper}}$ and $\mathcal{D}_{\text{lower}}$ have finite number of data points, it is easy to find $\alpha(\gamma)$ and $\beta(\gamma)$ such that $Q_{\text{upper}}(\alpha(\gamma)) = Q_{\text{lower}}(\beta(\gamma)) = 0$ up to the machine precision (See Section 3.2.1 for details). Then, the final $K$ PIs for the confidence levels $\Gamma = \{\gamma_k\}_{k=1}^K$ is given by

$$\big[ L(\boldsymbol{x}|\beta(\gamma_k)), U(\boldsymbol{x}|\alpha(\gamma_k)) \big] = \big[ f_{\boldsymbol{\omega}}(\boldsymbol{x}) + \nu - \beta(\gamma_k)l_{\boldsymbol{\xi}}(\boldsymbol{x}), f_{\boldsymbol{\omega}_k}(\boldsymbol{x}) + \nu + \alpha(\gamma_k)u_{\boldsymbol{\theta}}(\boldsymbol{x}) \big]. \tag{11}$$

**Remark 1** (Generate PIs for multiple $\gamma$ without re-training the NNs). *Note that the NN training in* Step 1-3 *is independent of the confidence level $\gamma$, so the PIs for multiple $\gamma$ can be obtained by only performing the root finding algorithm in* Step 4 *without re-training the three NNs. In contrast, most existing PI methods (e.g., QD and PIVEN) require re-training NNs when $\gamma$ is changed. This feature, ultimately enabled by Lemma 1, makes our method more efficient and convenient to use in practice.*

### 3.2.1 THEORETICAL ANALYSIS ON THE NON-CROSSING PROPERTY

The goal of this subsection is to theoretically prove that our algorithm can completely avoid the "crossing issue" [18] suffered by most existing PI methods without adding any non-crossing constraints to the loss function. To proceed, we first prove the existence of the roots of the functions in Eq. (10).

**Lemma 2** (Existence of the roots). *Given the trained models $f_{\boldsymbol{\omega}}(\boldsymbol{x})$, $u_{\boldsymbol{\theta}}(\boldsymbol{x})$, $l_{\boldsymbol{\xi}}(\boldsymbol{x})$ and $\nu$, if the training set $\mathcal{D}_{\text{train}}$ does not have repeated samples (i.e., $(\boldsymbol{x}_i, y_i) \neq (\boldsymbol{x}_j, y_j)$ if $i \neq j$), then, for any fixed $\gamma > 0$, there exist $\alpha(\gamma)$ and $\beta(\gamma)$ such that $Q_{\text{upper}}(\alpha(\gamma)) = Q_{\text{lower}}(\beta(\gamma)) = 0$.*

The proof of this lemma is given in Appendix B. In practice, the reason why it is easy to find the roots of $Q_{\text{upper}}$ and $Q_{\text{lower}}$ is that the number of training samples are finite, so that the bisection method can vary $\alpha$ and $\beta$ *continuously* in $[0, \infty)$ to search for "gaps" among the data points to find the zeros of $Q_{\text{upper}}(\alpha)$ and $Q_{\text{lower}}(\beta)$. Below is the theorem about the non-crossing property of the PIs obtained by our method. Lemma 2 will be used to in the proof of the theorem.

**Theorem 1** (Non-crossing property). *Given the trained models $f_{\boldsymbol{\omega}}(\boldsymbol{x})$, $u_{\boldsymbol{\theta}}(\boldsymbol{x})$, $l_{\boldsymbol{\xi}}(\boldsymbol{x})$ and $\nu$, if $\gamma > \gamma'$ satisfying $\lceil N(1-\gamma)/2 \rceil < \lceil N(1-\gamma')/2 \rceil$, then $U(\boldsymbol{x}|\alpha(\gamma)) > U(\boldsymbol{x}|\alpha(\gamma'))$ and $L(\boldsymbol{x}|\beta(\gamma)) < L(\boldsymbol{x}|\beta(\gamma'))$.*

The proof of this theorem is given in Appendix C. Theorem 1 tells us that the width of the PI obtained by our method will monotonically increasing with $\gamma$, which completely avoids the crossing issue. Again, this property is ultimately enabled by the equivalent representation of the PI in Lemma 1.

### 3.3 IDENTIFYING OUT-OF-DISTRIBUTION (OOD) SAMPLES

When using the trained model $f_{\boldsymbol{\omega}}(\boldsymbol{x})$ to make predictions for $\boldsymbol{x} \notin \mathcal{D}_{\text{train}}$, it is required that a UQ method can identify OOD samples and quantify their uncertainties, i.e., for $\boldsymbol{x} \notin \mathcal{D}_{\text{train}}$, the PI's width increases with the distance between $\boldsymbol{x}$ and $\mathcal{D}_{\text{train}}$. Inspired by the expressivity of ReLU networks, we add the OOD identification feature to the PI3NN method by initializing the bias of the output layer of $u_{\boldsymbol{\theta}}(\boldsymbol{x})$ and $l_{\boldsymbol{\xi}}(\boldsymbol{x})$ to a relatively large value. Specifically, when the OOD identification feature is turned on, we perform the following initialization before training $u_{\boldsymbol{\theta}}(\boldsymbol{x})$ and $l_{\boldsymbol{\xi}}(\boldsymbol{x})$ in *Step 3*.

- Pretrain $u_{\boldsymbol{\theta}}$ and $l_{\boldsymbol{\xi}}$ using the training set $\mathcal{D}_{\text{train}}$ with default initialization and compute the mean outputs $\mu_{\text{upper}} = \sum_{i=1}^N u_{\boldsymbol{\theta}}(\boldsymbol{x}_i)/N$ and $\mu_{\text{lower}} = \sum_{i=1}^N l_{\boldsymbol{\xi}}(\boldsymbol{x}_i)/N$ based on the training set.

- Initialize the biases of the output layers of $u_{\boldsymbol{\theta}}(\boldsymbol{x})$ and $l_{\boldsymbol{\xi}}(\boldsymbol{x})$ to $c\,\mu_{\text{upper}}$ and $c\,\mu_{\text{lower}}$, where $c$ is a large number (e.g., $c = 10$ in this work), such that the initial outputs of $u_{\boldsymbol{\theta}}(\boldsymbol{x})$ and $l_{\boldsymbol{\xi}}(\boldsymbol{x})$ are significantly larger than $\mu_{\text{upper}}$ and $\mu$.

- Re-train $u_{\boldsymbol{\theta}}(\boldsymbol{x})$ and $l_{\boldsymbol{\xi}}(\boldsymbol{x})$ following *Step 3* using the MSE loss.

The key in above initialization scheme is the increase of the bias of the output layer of $u_{\boldsymbol{\theta}}(\boldsymbol{x})$ and $l_{\boldsymbol{\xi}}(\boldsymbol{x})$. It is known that a ReLU network provides a piecewise linear function. The weights and biases of hidden layers defines how the input space is partitioned into a set of linear regions [25]; the weights of the output layer determines how those linear regions are combined; and the bias of the output layer acts as a shifting parameter. The weights and biases are usually initialized with some standard distribution, e.g., uniform or Gaussian. Setting the biases to $c\mu_{\text{upper}}$ and $c\mu_{\text{lower}}$ with a large value of $c$ will significantly increase the output of the initial $u_{\boldsymbol{\theta}}(\boldsymbol{x})$ and $l_{\boldsymbol{\xi}}(\boldsymbol{x})$ (See Figure 1 for demonstration). During the training, the loss in Eq. (8) will encourage the decrease of $u_{\boldsymbol{\theta}}(\boldsymbol{x})$ and $l_{\boldsymbol{\xi}}(\boldsymbol{x})$ only for the in-distribution (InD) samples (i.e., $\boldsymbol{x} \in \mathcal{D}_{\text{train}}$), not for the OOD samples. Therefore, after training, the PI will be wider in the OOD region than in the InD region. Additionally, due to continuity of the ReLU network, the PI width (PIW) will increase with the distance between $\boldsymbol{x}$ and $\mathcal{D}_{\text{train}}$ showing a decreasing confidence. Moreover, we can define the following *confidence score* by exploiting the width of the PI, i.e.,

$$\Lambda(\boldsymbol{x}) = \min \left\{ \frac{\sum_{i=1}^{N}(U(\boldsymbol{x}_i|\alpha(\gamma)) - L(\boldsymbol{x}_i|\beta(\gamma))/N}{U(\boldsymbol{x}|\alpha(\gamma)) - L(\boldsymbol{x}|\beta(\gamma))}, 1.0 \right\}, \tag{12}$$

where the numerator is the mean PI width (MPIW) of the training set $\mathcal{D}_{\text{train}}$ and the denominator is the PIW of a testing sample $\boldsymbol{x}$. If $\boldsymbol{x}$ is an InD sample, its confidence score should be close to one. As $\boldsymbol{x}$ moves away from $\mathcal{D}_{\text{train}}$, the PIW gets larger and thus the confidence score becomes smaller.

### 3.4 An illustrative example

We use a one-dimensional non-Gaussian cubic regression dataset to illustrate PI3NN. We train models on $y = x^3 + \varepsilon$ within $[-4, 4]$ and test them on $[-7, 7]$. The noise $\varepsilon$ is defined by $\varepsilon = s(\zeta)\zeta$, where $\zeta \sim \mathcal{N}(0, 1)$, $s(\zeta) = 30$ for $\zeta \geq 0$ and $s(\zeta) = 10$ for $\zeta < 0$. Top panels of Figure 1 illustrate the four steps of the PI3NN algorithm. After *Step 1-3* we finish the NNs training and obtain $f_{\boldsymbol{\omega}}(\boldsymbol{x}) + \nu - l_{\boldsymbol{\xi}}(\boldsymbol{x})$ and $f_{\boldsymbol{\omega}}(\boldsymbol{x}) + \nu + u_{\boldsymbol{\theta}}(\boldsymbol{x})$. Then for a given series of confidence levels $\gamma$, we use root-finding technique to calculate the corresponding $\alpha$ and $\beta$ in *Step 4* and obtain the associated $100\gamma\%$ PIs defined in Eq. (11). Figure 1 shows the 90%, 95% and 99% PIs. In calculation of the series of PIs for the multiple confidence levels, PI3NN only trains the three NNs once and the resulted PIs have no "crossing issue". Bottom panels of Figure 1 demonstrate the effectiveness of PI3NN's OOD identification capability and illustrate that it is our bias initialization scheme (Section 3.3) enables PI3NN to identify the OOD regions and reasonably quantify their uncertainty.

## 4 Experimental evaluation

We evaluate our PI3NN method by comparing its performance to four top-performing baselines—QD [14], PIVEN [15], SQR [17], and DER [6] on four experiments. The first experiment focuses on evaluation of the accuracy and robustness of the methods in calculating PIs. The last three experiments focus on the assessment of the OOD identification capability. The code of PI3NN is avaliable at https://github.com/liusiyan/PI3NN.

### 4.1 UCI regression benchmarks

We first evaluate PI3NN to calculate 95% PIs on nine UCI datasets [26]. The quality of PI is measured by the Prediction Interval Coverage Probability (PICP) which represents the ratio of data samples that fall within their respective PIs. We are interested in calculating a well-calibrated PIs, i.e., the test PICP is close to the desired value of 0.95 and meanwhile has a narrow width. In this experiment, we focus on the metric of PICP, not the PI width (PIW) because it is meaningful to compare PIWs only when the methods have the same PICPs. A small PICP certainly produces small PIWs, but the small PICP may be much lower than the desired confidence level causing unreasonable PIs. In this experiment, for different model settings, some PI methods produce a wide range of PICPs while

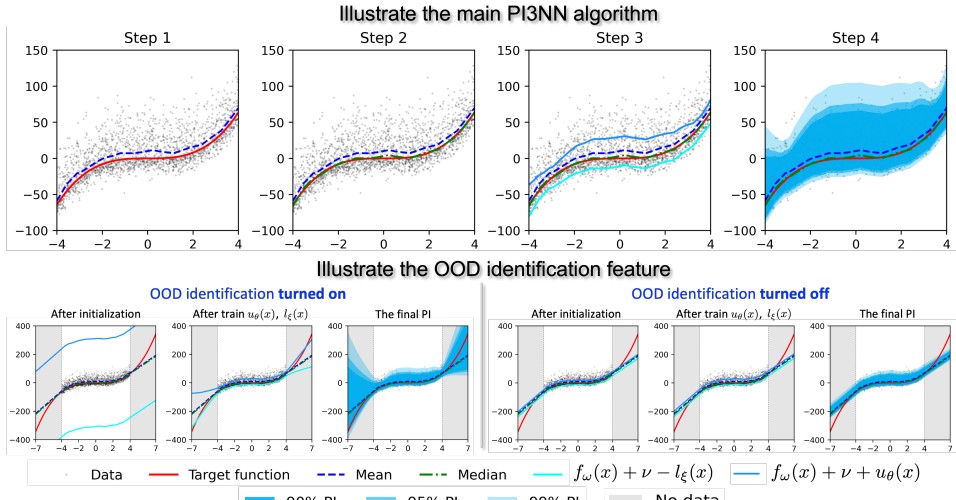

**Figure 1:** Top panels illustrate the four steps of our PI3NN algorithm. Bottom panels illustrate the effectiveness of the OOD identification feature. As shown in bottom left, when turning on the OOD identification feature by initializing the bias of the output layer of $u_{\boldsymbol{\theta}}$ and $l_{\boldsymbol{\xi}}$ to a large value, PI3NN can accurately identify the OOD regions $[-7, -4] \cup [4, 7]$ by giving them increasingly large PIs as their distance from the training data gets large. In bottom right, if we turn off the OOD identification by using the default initialization, PI3NN will not identify the OOD regions by giving them a narrow uncertainty bound.

others result in a relatively narrow range; it is unfair to compare the PIWs between methods that produce different ranges of PICPs.

We use a single hidden layer ReLU NN for all the methods. For each method, we test multiple scenarios with different hyper-parameter configurations, different training-testing data splits. For each scenario, we use the average of 5 runs (with 5 random seeds) to generate the result. All the methods consider the variation of the hidden layer size. QD and PIVEN additionally consider the variation of the extra hyper-parameters introduced in their loss functions. See Appendix for details about the experimental setup.

Table 1 summarizes the results. It shows the mean, standard deviation (std), best, and worst PICP (across hyper-parameter configurations) for all the methods on the testing datasets. DER, because of the Gaussian assumption, tends to overestimate the PI resulting in PICPs close to 1.0 for most datasets and produces the worst performance in terms of PICP. For the four PI methods, our PI3NN achieves the best mean PICP (closest to 0.95) with the smallest std. PI3NN is also the top performer in terms of the best and the worst PICP. This suggests that PI3NN can produce a well-calibrated PI and its performance is less affected by the hyper-parameter configuration. In contrast, QD, PIVEN, and SQR require a careful hyper-parameter tuning to obtain the comparable best PICP as the PI3NN, but for each hyper-parameter set, their PICPs vary a lot with a large std and the worst PICP can be much lower than the desired 0.95 resulting in an unreasonable PI. Thus, this experiment demonstrates the advantage of our method in producing a well-calibrated PI without time-consuming hyperparameter fine tuning, which makes it convenient for practical use.

## 4.2 OOD IDENTIFICATION IN A 10-DIMENSIONAL FUNCTION EXAMPLE

We use a 10-dimensional cubic function $f(\boldsymbol{x}) = \frac{1}{10}(x_1^3 + \cdots + x_{10}^3)$ to demonstrate the effectiveness of the proposed bias initialization strategy in Section 3.3 for OOD identification. The training data of $\boldsymbol{x}$ is generated by drawing 5,000 samples from the normal distribution $\mathcal{N}(0, 1)$; the corresponding training data of outputs $y$ is obtained from $y = f(\boldsymbol{x}) + \varepsilon$ with $\varepsilon$ following a Gaussian distribution $\mathcal{N}(0, 1)$. We define a test set with 1,000 OOD samples, where $\boldsymbol{x}$ are drawn from a shifted normal distribution $\mathcal{N}(2, 1)$. We calculate the 90% PI for both the training (InD) and the testing (OOD) data using all the five methods. For PI3NN, we consider two situations with the OOD identification turned on and turned off. We set the constant $c$ in Section 3.3 to 10 to turn on PI3NN's OOD identification feature, and use the standard initialization to turn the feature off. We use PI width (PIW) to evaluate

|  |  | Boston | Concrete | Energy | Kin8nm | Naval | Power | Protein | Wine | Yacht |
|---|---|---|---|---|---|---|---|---|---|---|
| PI3NN | Mean | 0.95 | 0.94 | 0.95 | 0.94 | 0.95 | 0.95 | 0.95 | 0.95 | 0.95 |
|  | Std | 0.03 | 0.02 | 0.03 | 0.006 | 0.006 | 0.004 | 0.003 | 0.02 | 0.02 |
|  | Best | 0.94 | 0.95 | 0.95 | 0.95 | 0.95 | 0.95 | 0.95 | 0.95 | 0.94 |
|  | Worst | 0.88 | 0.89 | 0.87 | 0.93 | 0.94 | 0.96 | 0.94 | 0.91 | 0.90 |
| QD | Mean | 0.85 | 0.84 | 0.87 | 0.91 | 0.94 | 0.94 | 0.93 | 0.90 | 0.93 |
|  | Std | 0.06 | 0.05 | 0.04 | 0.02 | 0.03 | 0.01 | 0.02 | 0.03 | 0.06 |
|  | Best | 0.94 | 0.95 | 0.95 | 0.95 | 0.95 | 0.95 | 0.95 | 0.95 | 0.93 |
|  | Worst | 0.69 | 0.66 | 0.78 | 0.86 | 0.59 | 0.92 | 0.90 | 0.83 | 0.71 |
| PIVEN | Mean | 0.83 | 0.83 | 0.82 | 0.87 | 0.94 | 0.94 | 0.91 | 0.82 | 0.88 |
|  | Std | 0.07 | 0.05 | 0.05 | 0.02 | 0.01 | 0.01 | 0.02 | 0.04 | 0.08 |
|  | Best | 0.94 | 0.94 | 0.93 | 0.91 | 0.95 | 0.95 | 0.94 | 0.91 | 0.93 |
|  | Worst | 0.51 | 0.50 | 0.67 | 0.82 | 0.91 | 0.90 | 0.88 | 0.68 | 0.52 |
| SQR | Mean | 0.76 | 0.83 | 0.83 | 0.86 | 0.87 | 0.88 | 0.87 | 0.85 | 0.82 |
|  | Std | 0.17 | 0.12 | 0.11 | 0.13 | 0.14 | 0.13 | 0.13 | 0.12 | 0.13 |
|  | Best | 0.94 | 0.95 | 0.96 | 0.95 | 0.95 | 0.95 | 0.95 | 0.95 | 0.94 |
|  | Worst | 0.39 | 0.52 | 0.58 | 0.56 | 0.54 | 0.56 | 0.51 | 0.42 | 0.48 |
| DER | Mean | 0.87 | 1.0 | 0.98 | 1.0 | 1.0 | 1.0 | 1.0 | 0.98 | 0.83 |
|  | Std | 0.03 | 0.0 | 0.04 | 0.0 | 0.0 | 0.0 | 0.004 | 0.008 | 0.1 |
|  | Best | 0.94 | 1.0 | 0.94 | 1.0 | 1.0 | 1.0 | 0.98 | 0.97 | 0.93 |
|  | Worst | 0.80 | 1.0 | 1.0 | 1.0 | 1.0 | 1.0 | 1.0 | 1.0 | 0.61 |

**Table 1:** Evaluation of 95% PI on testing data. We show the mean, standard deviation (std), best, and worst PICP (across hyper-parameter configurations) for all methods. The best performer should produce PICP values closest to the desired 0.95. DER tends to overestimate the PI resulting in PICPs close to 1.0 for most datasets and produces the worst performance. For the four PI methods, our PI3NN shows the top performance by giving the best mean PICP (closest to 0.95) with the smallest std across hyper-parameter configurations. QD, PIVEN, and SQR require a careful hyper-parameter tuning to obtain the comparable best PICP as the PI3NN, but for each hyper-parameter set, their PICP std is large and the worst PICP can be much lower than the desired 0.95.

the method's capability in OOD identification. A method having an OOD identification capability should produce larger PIWs of the OOD samples than those of the InD samples. Additionally, we use the confidence score defined in Eq. (12) to quantitatively evaluate the method's capability in OOD identification. InD samples should have a close-to-one confidence score and the OOD samples should have a remarkably smaller confidence score than that of the InD datasets.

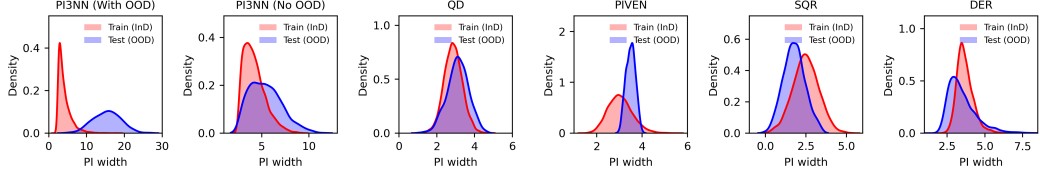

**Figure 2:** Probability density functions of the PI width for the training (InD) and testing (OOD) data for all the five methods. When we use a probability density function (PDF) to fit PIWs of the InD and OOD samples, respectively, we should be able to see two separated PDFs with the PDF of OOD samples shifting to the right having larger PIWs. If the two PDFs are overlapped to each other, then it indicates the method can not identify the OOD samples by reasonably quantifying their uncertainty.

|  | PI3NN (With OOD) | PI3NN (No OOD) | QD | PIVEN | SQR | DER |
|---|---|---|---|---|---|---|
| Train (InD) | $0.91 \pm 0.15$ | $0.92 \pm 0.12$ | $0.94 \pm 0.08$ | $0.94 \pm 0.08$ | $0.91 \pm 0.12$ | $0.95 \pm 0.08$ |
| Test (OOD) | $0.28 \pm 0.08$ | $0.80 \pm 0.18$ | $0.90 \pm 0.10$ | $0.87 \pm 0.05$ | $0.98 \pm 0.07$ | $0.94 \pm 0.11$ |

**Table 2:** Mean $\pm$ Standard Deviation of the confidence score (Eq. (12) with $\gamma = 0.9$) for the 10D cubic example

Figure 2 shows the PDFs of the PIWs of the InD and OOD samples. The figures indicate that our PI3NN method, with OOD identification feature turned on, can effectively identify the OOD samples by assigning them larger and well-separated PIWs than those of the InD dataset. The other methods are not able to identify the OOD samples by mixing their PIWs into the InD data's PIWs, showing

overlapped PDFs. These methods' performance is similar to the case when the PI3NN turns off its OOD identification capability. Table 2 lists the mean and std of the confidence scores for the InD and OOD datasets. This table further shows that PI3NN with OOD identification feature turned on can effectively identify the OOD samples by giving them smaller confidence scores than those of the InD samples, while other methods produce over-confident (wider) PIs for the OOD samples and cannot separate OOD samples from InD dataset.

### 4.3 OOD IDENTIFICATION IN A FLIGHT DELAY DATASET

We test the performance of our method in OOD detection using the Flight Delay dataset [27], which contains about 2 million US domestic flights in 2008 and their cases for delay. We use the PI3NN method to predict the regression map from 6 input parameters, i.e., Day of Month, Day of Week, AirTime, Flight Distance, TaxiIn (taxi in time), and TaxiOut (taxi out time), to the Arrival Delay. The US federal aviation administration (FAA) categories general commercial airports into several ranks, i.e., *Rank 1 (Large Hub)*: 1% or more of the total departures per airport, *Rank 2 (Medium Hub)*: 0.25% - 1.0% of the total departures per airport, *Rank 3 (Small Hub)*: 0.05% - 0.25% of the total departures per airport, and *Rank 4 (NonHub)*: <0.05% but more than 100 flights per year.

The training set consists of randomly select 18K data from Rank 4 (Nonhub) airports' data. We also randomly select another 20K data from each rank to form 4 testing sets. It is known that there is a data shift from Rank 1 airports to Rank 4 airports, because the bigger an airport, the longer the Taxi time, the averaged AirTime and the Flight Distance. Thus, an OOD-aware PI method should be able to identify those data shift. The setup for PI3NN method and the baselines are given in Appendix.

Table 3 shows the confidence scores (i.e., Eq. (12) with $\gamma = 0.9$) computed by our method and the baselines. Our method with the OOD identification feature turned on successfully detected the data shift by giving the Rank 1,2,3 testing data lower confidence scores. In contrast, other methods as well as our method (with OOD identification turned off) fail to identify the data shift from the training set to the Rank 1,2,3 testing sets.

| Data | Rank 1 | Rank 2 | Rank 3 | Rank 4 |
|------|--------|--------|--------|--------|
| PI3NN (With OOD) | $0.24 \pm 0.05$ | $0.32 \pm 0.09$ | $0.72 \pm 0.22$ | $0.92 \pm 0.24$ |
| PI3NN (No OOD) | $0.72 \pm 0.32$ | $0.79 \pm 0.32$ | $0.88 \pm 0.27$ | $0.90 \pm 0.26$ |
| QD | $0.89 \pm 0.24$ | $0.90 \pm 0.24$ | $0.92 \pm 0.16$ | $0.83 \pm 0.14$ |
| PIVEN | $0.83 \pm 0.23$ | $0.87 \pm 0.21$ | $0.99 \pm 0.04$ | $0.99 \pm 0.03$ |
| SQR | $0.98 \pm 0.06$ | $0.96 \pm 1.41$ | $0.97 \pm 0.09$ | $0.94 \pm 0.10$ |
| DER | $0.98 \pm 0.13$ | $1.00 \pm 0.01$ | $1.00 \pm 0.00$ | $1.00 \pm 0.01$ |

**Table 3:** Mean $\pm$ Standard Deviation of the confidence score (Eq. (12) with $\gamma = 0.9$) for the flight delay data.

## 5 CONCLUSION AND DISCUSSION

The limitations of the PI3NN method include: (1) For a target function with multiple outputs, each output needs to have its own PI and OOD confidence score. The PI and the confidence score cannot oversee all the outputs. For example, this could make it challenging to apply PI3NN to NN models having image as outputs, (e.g., autoencoders). (2) The effectiveness of the OOD detection approach depends on that there are sufficiently many piecewise linear regions (of ReLU networks) in the OOD area. So far, this is achieved by the standard random initialization (ensure uniform distributed piecewise linear regions at the beginning of training) and $L_1/L_2$ regularization (ensure the linear regions not collapse together around the training set). However, there is no guarantee of uniformly distributed piecewise linear regions after training. Improvement of this requires significant theoretical work on how to manipulate the piecewise linear function defined by the ReLU network. This work is for purely research purpose and will have no negative social impact. (3) When the noise $\varepsilon$ has heavy tails, our method will struggle when the distance between tail quantiles becomes very large. In this case, the number of training data needed to accurately capture those tail quntiles may be too large to afford. It is a common issue for all distribution-free PI methods, and reasonable distributional assumption may be needed to alleviate this issue.

## 6 ACKNOWLEDGEMENT

This work was supported by the U.S. Department of Energy, Office of Science, Office of Advanced Scientific Computing Research, Applied Mathematics program; and by the Artificial Intelligence Initiative at the Oak Ridge National Laboratory (ORNL). ORNL is operated by UT-Battelle, LLC., for the U.S. Department of Energy under Contract DE-AC05-00OR22725. This manuscript has been authored by UT-Battelle, LLC. The US government retains and the publisher, by accepting the article for publication, acknowledges that the US government retains a nonexclusive, paid-up, irrevocable, worldwide license to publish or reproduce the published form of this manuscript, or allow others to do so, for US government purposes. DOE will provide public access to these results of federally sponsored research in accordance with the DOE Public Access Plan (http://energy.gov/downloads/doe-public-access-plan). We would like to thank Paul Laiu at ORNL for helpful discussions.

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

APPENDIX

## A  THE PROOF OF LEMMA 1

*Proof.* Substitute $y = f(\boldsymbol{x}) + \varepsilon$ into Eq. (5), we can rewrite Eq. (5) as $\hat{U}(\boldsymbol{x}|\alpha) = f(\boldsymbol{x}) + \mathbb{M}[\varepsilon] + \alpha\mathbb{E}[(\varepsilon - \mathbb{M}[\varepsilon])\mathbf{1}_{\varepsilon-\mathbb{M}[\varepsilon]>0}]$, where $\mathbb{E}[(\varepsilon - \mathbb{M}[\varepsilon])\mathbf{1}_{\varepsilon-\mathbb{M}[\varepsilon]>0}] = \mathbb{E}[(y - \mathbb{M}[y])\mathbf{1}_{y-\mathbb{M}[y]>0}]$ is strictly positive and independent of $\boldsymbol{x}$. Then, we have that $\hat{U}(\boldsymbol{x}|\alpha)$ satisfies the property (P1).

For a fixed $\boldsymbol{x}$, there exists a unique value $\tilde{\alpha}(\gamma, \boldsymbol{x})$, which depends on both $\gamma$ and $\boldsymbol{x}$, such that

$$\mathbb{E}[\mathbf{1}_{y>\hat{U}(\boldsymbol{x}|\tilde{\alpha}(\gamma,\boldsymbol{x}))}] - (1 - \gamma)/2 = 0. \tag{13}$$

Comparing Eq. (13) and the definition of the upper bound in Eq. (3), we can see that $\hat{U}(\boldsymbol{x}|\tilde{\alpha}(\gamma, \boldsymbol{x}))$ is also a upper bound of the $100\gamma\%$ PI for a given $\boldsymbol{x}$. Due to the uniqueness of the upper bound, i.e., the property (P3), we have $\hat{U}(\boldsymbol{x}|\tilde{\alpha}(\gamma, \boldsymbol{x})) = U_\gamma^{\text{true}}(\boldsymbol{x})$.

Because $\hat{U}(\boldsymbol{x}|\tilde{\alpha}(\gamma, \boldsymbol{x}))$ is the upper bound of the $100\gamma\%$ PI, it satisfies the property (P2), i.e., $\hat{U}(\boldsymbol{x}|\tilde{\alpha}(\gamma, \boldsymbol{x})) - f(\boldsymbol{x})$ is independent of $\boldsymbol{x}$. Then we have

$$\hat{U}(\boldsymbol{x}|\tilde{\alpha}(\gamma, \boldsymbol{x})) - f(\boldsymbol{x}) = \mathbb{M}[\varepsilon] + \tilde{\alpha}(\gamma, \boldsymbol{x})\mathbb{E}[(\varepsilon - \mathbb{M}[\varepsilon])\mathbf{1}_{\varepsilon-\mathbb{M}[\varepsilon]>0}]$$

is independent of $\boldsymbol{x}$. Since both $\mathbb{M}[\varepsilon]$ and $\mathbb{E}[(\varepsilon - \mathbb{M}[\varepsilon])\mathbf{1}_{\varepsilon-\mathbb{M}[\varepsilon]>0}]$ are independent of $\boldsymbol{x}$, we have $\alpha(\gamma) := \tilde{\alpha}(\gamma, \boldsymbol{x})$ is also independent of $\boldsymbol{x}$, which concludes the proof.

$\square$

## B  THE PROOF OF LEMMA 2

*Proof.* We only need to prove $Q_{\text{upper}}(\alpha(\gamma)) = 0$, and the same derivation can be applied to $Q_{\text{lower}}(\beta(\gamma))$. Also, for notational simplicity, we assume $N$ is an even number, so that $N/2$ is an integer. We first define

$$g(\alpha) = \sum_{(\boldsymbol{x}_i,y_i)\in\mathcal{D}_{\text{upper}}} \mathbf{1}_{y_i>U(\boldsymbol{x}_i|\alpha)}, \quad \text{for } \alpha \in [0, \infty), \tag{14}$$

based on the definition of $Q_{\text{upper}}$ in Eq. (10). Since $N$ is a finite integer, $g(\alpha)$ can be written as a step function, i.e.,

$$g(\alpha) = \sum_{k=0}^{N/2} k\, \mathbf{1}_{A_k}(\alpha), \tag{15}$$

where $\mathbf{1}_{A_k}(\alpha)$ is defined by

$$\mathbf{1}_{A_k}(\alpha) = \begin{cases} 1, & \text{if } \alpha \in A_k, \\ 0, & \text{if } \alpha \notin A_k, \end{cases} \tag{16}$$

with $A_k$ defined by

$$A_k = \left\{\alpha \in [0, \infty) \,\Big|\, \#\{y_i > U(\boldsymbol{x}_i|\alpha) \text{ for } (\boldsymbol{x}_i, y_i) \in \mathcal{D}_{\text{upper}}\} = k\right\}.$$

In other words, $A_k$ is a sub-interval of $[0, \infty)$, such that the number of samples in $\mathcal{D}_{\text{upper}}$ satisfying $y_i > U(\boldsymbol{x}_i|\alpha)$ is equal to $k$ for $\alpha \in A_k$.

Since $(\boldsymbol{x}_i, y_i) \neq (\boldsymbol{x}_j, y_j)$ for $i \neq j$, the Lebesgue measure of $A_k$, denoted by $\lambda(A_k)$ is strictly positive, i.e., $\lambda(A_k) > 0$. From classic measure theory, we know that for any set in $\mathbb{R}$ with strictly positive Lebesgue measure, there are *infinite* real numbers in the set. As such, there are infinite real numbers in each $A_k$ for $k = 0, \ldots, N/2$.

Hence, for a given $\gamma$, any real number $\alpha(\gamma) \in A_{\lceil N(1-\gamma)/2\rceil}$ will satisfy that $g(\alpha(\gamma)) = \lceil N(1-\gamma)/2\rceil$, i.e., $Q_{\text{upper}}(\alpha(\gamma)) = 0$, which concludes the proof. $\square$

## C    THE PROOF OF THEOREM 1

*Proof.* We only need to prove $U(\boldsymbol{x}|\alpha(\gamma)) > U(\boldsymbol{x}|\alpha(\gamma'))$, and the same derivation can be applied to $L(\boldsymbol{x}|\beta)$. Since $u_{\boldsymbol{\theta}}(\boldsymbol{x}) \geq 0$, we have that $U(\boldsymbol{x}|\alpha)$ is monotonically increasing with $\alpha$ for any $\boldsymbol{x}$. Thus, if $\alpha(\gamma) > \alpha(\gamma')$ then $U(\boldsymbol{x}|\alpha(\gamma)) > U(\boldsymbol{x}|\alpha(\gamma'))$. So we only need to prove $\alpha(\gamma) > \alpha(\gamma')$. We use the proof by contradiction approach to prove $\alpha(\gamma) > \alpha(\gamma')$, i.e., we will derive a contradiction from the assumption that $\alpha(\gamma) \leq \alpha(\gamma')$.

It is easy to see that the function

$$g(\alpha) = \sum_{(\boldsymbol{x}_i, y_i) \in \mathcal{D}_{\text{upper}}} \mathbf{1}_{y_i > U(\boldsymbol{x}_i|\alpha)}, \quad \text{for } \alpha \in [0, \infty), \tag{17}$$

is a monotonically decreasing function of $\alpha$. Thus, if $\alpha(\gamma) \leq \alpha(\gamma')$, we have $g(\alpha(\gamma)) \geq g(\alpha(\gamma'))$. On the other hand, we know from Lemma 2 that $\alpha(\gamma)$ and $\alpha(\gamma')$ satisfy

$$g(\alpha(\gamma)) = \lceil N(1-\gamma)/2 \rceil \quad \text{and} \quad g(\alpha(\gamma')) = \lceil N(1-\gamma')/2 \rceil, \tag{18}$$

which leads to $\lceil N(1-\gamma)/2 \rceil \geq \lceil N(1-\gamma')/2 \rceil$. This is a contradiction with the condition of the theorem that $\lceil N(1-\gamma)/2 \rceil < \lceil N(1-\gamma')/2 \rceil$. Therefore, the assumption $\alpha(\gamma) \leq \alpha(\gamma')$ is incorrect, such that we have $\alpha(\gamma) > \alpha(\gamma') \Rightarrow U(\boldsymbol{x}|\alpha(\gamma)) > U(\boldsymbol{x}|\alpha(\gamma'))$, which concludes the proof.    $\square$

## D    EXPERIMENT SETUPS AND PARAMETERS

### D.1    SETUP FOR THE UCI EXAMPLES AND EXTRA RESULTS

We evaluated the performance of the five method on 9 widely used UCI data sets, including Boston housing (boston), Concrete compressive strength (concrete), Energy efficiency (energy), KINematics 8 inputs non-linear medium unpredictability/noise (kin8nm), Combined Cycle Power Plant (power), Physicochemical Properties of Protein Tertiary Structure (Protein), Wine quality (wine), and yacht Hydrodynamics (yacht). Since data splitting will affect the results for those problems, we pre-split the each data set with fixed splitting random seed, and generated 3 train/test (90%/10%) data pairs used for all methods, so as to ensure a fair comparison. Even different methods are built on top of different ML/DL platform, software versions, we intend to standardize the data pre-processing steps (e.g. data normalization) to ensure same data is imported to all models.

For each method, we test multiple scenarios with different hyper-parameter configurations, different training-testing data splits and random seeds. All the methods consider the variation of the hidden layer size. QD and PIVEN additionally consider the variation of the extra hyper-parameters introduced in their loss functions.

We summarized the modeling results in Table 1 of the main text. In below Table 4 we listed the mean PI width (MPIW) for the cases having similar PICPs. As discussed, it is meaningful to compare the PI width only when the methods have the same PICPs. So, for the methods having the similar PICPs (i.e., the best PICP values highlighted in Table 1), we compared their MPIWs; those methods giving the smaller MPIW perform better. Table 4 indicates that for the similar PICP values, our PI3NN methods gave the smallest MPIWs for most of the datasets (7 out of 9). Please note that PI3NN obtained small MPIWs using the standard MSE loss, while other PI methods such as QD and PIVEN specifically minimized MPIWs in their loss function. This customized loss although sometimes gives better MPIWs after careful hyper-parameter tuning, it also results in unreliable prediction performance sensitive to the hyper-parameter configuration (as discussed in Table 1).

In the following, we discuss the model setup for each method in detail.

We use a single hidden layer ReLU NN for all the methods. For each method, we test multiple scenarios with different hyper-parameter configurations, different training-testing data splits. Specifically, we generate 3 pre-split data pairs for all experiments. The other universally applied hyperparameter for all methods is the number of hidden neurons: [50, 100, 200]. QD and PIVEN additionally consider the variation of the extra hyper-parameters introduced in their loss functions. For each scenario, we use the average of 5 runs (with 5 random seeds) to generate the result.

For PI3NN method, we use the hyper-parameters for all experiments: learning rate (0.01), Adam optimizer with MSE loss for all three networks.

|  |  | Boston | Concrete | Energy | Kin8nm | Naval | Power | Protein | Wine | Yacht |
|---|---|---|---|---|---|---|---|---|---|---|
| PI3NN | PICP | 0.94 | 0.95 | 0.95 | 0.95 | 0.95 | 0.95 | 0.95 | 0.95 | 0.94 |
|  | MPIW | 0.26 | 0.31 | 0.17 | 0.26 | 0.15 | 0.2 | 0.8 | 0.47 | 0.05 |
| QD | PICP | 0.94 | 0.95 | 0.95 | 0.95 | 0.95 | 0.95 | 0.95 | 0.95 |  |
|  | MPIW | 0.26 | 0.22 | 0.23 | 0.44 | 0.97 | 0.2 | 0.8 | 0.42 |  |
| PIVEN | PICP | 0.94 |  |  |  | 0.95 | 0.95 |  |  |  |
|  | MPIW | 0.31 |  |  |  | 0.28 | 0.2 |  |  |  |
| SQR | PICP | 0.94 | 0.95 |  | 0.95 | 0.95 | 0.95 | 0.95 | 0.95 | 0.94 |
|  | MPIW | 0.54 | 0.64 |  | 0.64 | 0.99 | 0.55 | 0.83 | 0.52 | 0.66 |
| DER | PICP | 0.94 |  |  |  |  |  |  |  |  |
|  | MPIW | 0.27 |  |  |  |  |  |  |  |  |

**Table 4:** Evaluation of 95% PI on testing data. We compare the mean PI width (MPIW) between methods when they have the similar PICP values (i.e., the best PICP highlighted in Table 1). For the similar PICP, the method producing the smaller MPIW performs better. Our PI3NN shows the top performance by giving the smallest MPIW for 7 out of 9 datasets. QD and PIVEN can also produce small MPIW for some datasets, but they obtained the small MPIW by specifically minimizing the MPIW in their customized loss function which resulted in a prediction performance sensitive to the hyper-parameter configuration.

For QD method, we include additional two hyper-parameter combinations: soften parameter: [100., 160., 200.] and lambda_in parameter: [5., 15., 25.] which are essential parameters for QD method. In addition, we keep the default parameters in the original QD code for boston and concrete data sets, and our tuned parameters for rest of 7 data. Specifically, the maximum epochs for all data is 300 except for concrete (800); similarly, the learning rate for concrete is 0.03 and 0.02 for other data sets; the decay rate and sigma_in are set to 0.9 and 0.1 respectively for 7 data sets except for concrete (0.98 and 0.2). The outer layer bias are set to [3., -3]. Adam optimizer is used in the experiments, and we use 100 batch size from the original code.

For DER method, the maximum epochs is set to 40. We follow the author tuned learning rate and batch size provided in the original code. The learning rate and batch size are (1e-3, 8) for boston, (5e-3, 1) for concrete, (2e-3, 1) for energy, (1e-3, 1) for kin8nm, (1e-3, 2) for power, (5e-4, 1) for naval, (1e-4, 32) for wine, (1e-3, 64) for protein, and (5e-4, 1) for yacht.

For PIVEN method, similar to QD, we include the lambda_in ([5.0, 15.0, 20.0]) and sigma_in ([0.05, 0.2, 0.4]) for the hyper-parameter combinations. The rest of the parameters are provided by the original authors in the code, including: batch size (100), outer layer biases [3., -3.], soften parameter (160.0). Meanwhile, we use the author tuned maximum epochs, learning rate, decay rate for boston (300, 0.02, 0.9), concrete (800, 0.03, 0.98), energy (1200, 0.016, 0.97), kin8nm (800, 0.0134725, 0.993922), naval (1000, 0.006, 0.998), power (500, 0.014075, 0.987099), protein (600, 0.002, 0.999), wine (1000, 0.01, 0.98) and yacht (2000, 0.005, 0.98).

For SQR method, we added three hyper-parameter combinations based on the original paper and code, which are the learning rate [1e-2, 1e-3, 1e-4], dropout rate [0.1, 0.25, 0.5, 0.75] and weight decay [0, 1e-3, 1e-2, 1e-1, 1]. maximum epochs is 5,000 for all data sets by default. Additional 20% validation data is split from the training data. These parameter combinations with the split seed, random seed, number of neurons yield much more resutls than other methods, thus, we reduced the results data size by fixing the learning rate (1e-2), dropout rate (0.1) and weight decay (0.1) for the final evaluations.

## D.2 SETUP FOR THE 10D EXAMPLE

The 10-dimensional cubic function is used for generating the synthetic data for demonstrating OOD identification capability. As mentioned in Section 4.2, we generate a training data with 10 input features and size of 5,000, and 1,000 OOD testing data.

For PI3NN method, we use three networks with single layer (100 hidden nodes) with ReLU activation, the MSE loss is used with SGD optimizer (learning rate = 0.01). Maximum epochs is 3,000. We use comparable setups for the baseline methods. Specifically, we take the hyperparameter values (suggested by the authors of those methods) and perform hyperparameter tuning around those

suggested values, and choose the best performance for each baseline method. The hyperparameters we tune include the hidden layer width, the learning rate and the extra hyperparameters introduced into the baseline method. For hidden layers, we use 100,200,300 hidden neurons as candidates. For learning rates, we use 0.0001, 0.001, 0.01 as candidates. For the exclusive hyperparameters, we choose five candidate values (including the suggested ones). For each hyperparameter configuration, we run an ensemble of 5 runs to get the results. After tuning, we have the following setup for the baseline methods. For QD method, we use single hidden layer network (200 hidden nodes) with Adam optimizer (lr=0.02). Activation function is ReLU, and batch size is 100. The soften parameter, labmda_in, and sigma_in are set to 160.0, 15.0 and 0.4, respectively. For PIVEN method, sigma_in (0.2), learning rate (0.01). It shares the same network structure, softer parameter and lambda_in with QD method. For DER method, we use single hidden layer with 200 nodes network, 1e-4 learning rate, 256 batch size. For SQR method, we use single layer network with 100 neurons network. We fix the learning rate, dropout rate, weight decay rate to 1e-3, 0.1 and 0, respectively. We also use the default setting by taking 20% of the training data as validation set in this method.

### D.3 SETUP FOR THE FLIGHT DELAY EXAMPLE

The flight delay data (`www.kaggle.com/vikalpdongre/us-flights-data-2008`) contains the flight information in the U.S. in the year 2008. We separate the data by airports into 4 ranks based on the percentage of departure flights, which including: Large Hub (Rank 1), receives 1% or more of the annual departures; Medium Hub (Rank 2), which receives 0.25%-1% of the annual departures; Small Hub (Rank 3) and Nonhub (rank4) receives 0.05%-0.25% and <0.05% of the departures, respectively. We select 5% of the Rank 4 data as the training data, another 5% of the Rank 4 is picked as 'test 4' data, and we consider this as in-distribution data. Rest of three testing data ('test 3', 'test 2', and 'test 1') are selected from the Rank 3, 2 and 1, respectively with the 5% total amount of data. We pick the 6 input features ('DayofMonth', 'DayOfWeek', 'AirTime', 'Distance', 'DepDelay', 'TaxiOut'), and select 'ArrDelay' as the output feature.

For PI3NN method, in the experiment, we use the same network architecture (i.e., one hidden layer ReLU network containing 100 hidden neurons) and as the one used for the UCI datasets. The Adam optimizer is used with learning rate being 0.01 and the maximum epochs being 50,000. Conventional L1 and L2 regularization are implemented with both penalties set to 0.02. The scalar parameter (i.e. the one controlling the initial bias of the output layer) is set to 10 (the same as the test on the 10D function).

We also perform hyperparameter tuning for each of the baseline methods. The universal hyperparameter, i.e., the number of hidden neurons come from the pool [50,100,200]. For the exclusive hyperparameters, we choose five candidate values (including the suggested ones). For each hyperparameter configuration, we run an ensemble of 5 runs to get the results. After tuning, we have the following setup for the baseline methods.

For QD method, we use single layer network with 100 neurons, Adam optimizer with 0.02 learning rate, 50 epochs, 160.0 soften parameter, 0.1 sigma_in, 15.0 lambda_in, 100 batch size, 0.9 decay rate.

For DER method, we use single hidden layer with 100 neurons network structure, 512 batch size, 1e-4 learning rate and 100 maximum epochs for the experiments.

For PIVEN method, lambda_in and sigma_in are set to 15.0 and 0.2. Same single layer 100 nodes network structure is applied. Soften parameter is 160.0. Batch size and maximum epochs are 100 and 500, respectively. We use 0.01 learning rate and 0.99 decay rate

For SQR method, we use same network structure [100 neurons] and maximum epochs (500) with PIVEN method. We fixed the learning rate (1e-3), dropout rate (0.1), and weight decay rate (0) in the flight delay experiments.

### D.4 OOD IDENTIFICATION IN A WATERSHED STREAMFLOW DATASET

We test the performance of our method in OOD detection using the streamflow dataset measured in East River Watershed, Colorado, United States. The dataset contains three years of daily data of precipitation, maximum temperature, minimum temperature, and streamflow at the catchment outlet. We use the Long Short-Term Memory (LSTM) network to learn the relationship between the three

meteorological forcing (i.e., precipitation, maximum temperature and minimum temperature) and the streamflow. The training data are from the first two years (2015-2016) and the third year (2017) of data form the testing set. Note that 2017 is a wet year with a relatively large amount of precipitation, which causes the streamflow patterns in 2017 dramatically different from those in 2015-2016.

In this experiment, the LSTM network consists of two sub-networks, i.e., the recurrent layers and the dense layers. The recurrent layers extract the temporal feature information and the dense layers learn the rainfall-runoff relationship. The prediction uncertainty quantification focuses on the dense layer training. In implementation, we first train the entire LSTM network to get the mean and the median prediction following Step 1 and 2 in Section 3.2. Next, we keep the recurrent layers unchanged and extract their outputs for all the LSTM input training samples, and use these output samples as inputs together with the streamflow data to train the dense layers and apply the UQ method to quantify the prediction uncertainty. For PI3NN, we use these samples to train $u_{\boldsymbol{\theta}}$ and $l_{\boldsymbol{\xi}}$ as mappings from the feature space (i.e., the input of $u_{\boldsymbol{\theta}}$ and $l_{\boldsymbol{\xi}}$ is the input of the dense layers of the LSTM network) to the output of the LSTM network. For other UQ methods such as QD, PIVEN, SQR, and DER, we use these samples to train the dense layers directly to obtain the uncertainty estimate. In this way, we make a fair comparison between PI3NN and the baselines where they all use the same calibrated recurrent layers and focus on the training of the dense layers.

Figure 3 shows the predicted value and its 95% PI. PI3NN can accurately identify the OOD data (i.e., the extreme event in 2017) by giving their predictions a large PI, while other methods fail to identify the OOD patterns by giving the testing data similarly narrow PIs as the training data. This example demonstrates that our PI3NN method can show a consistent superior performance for a different dataset (time-series data) and a different network architecture (LSTM network).

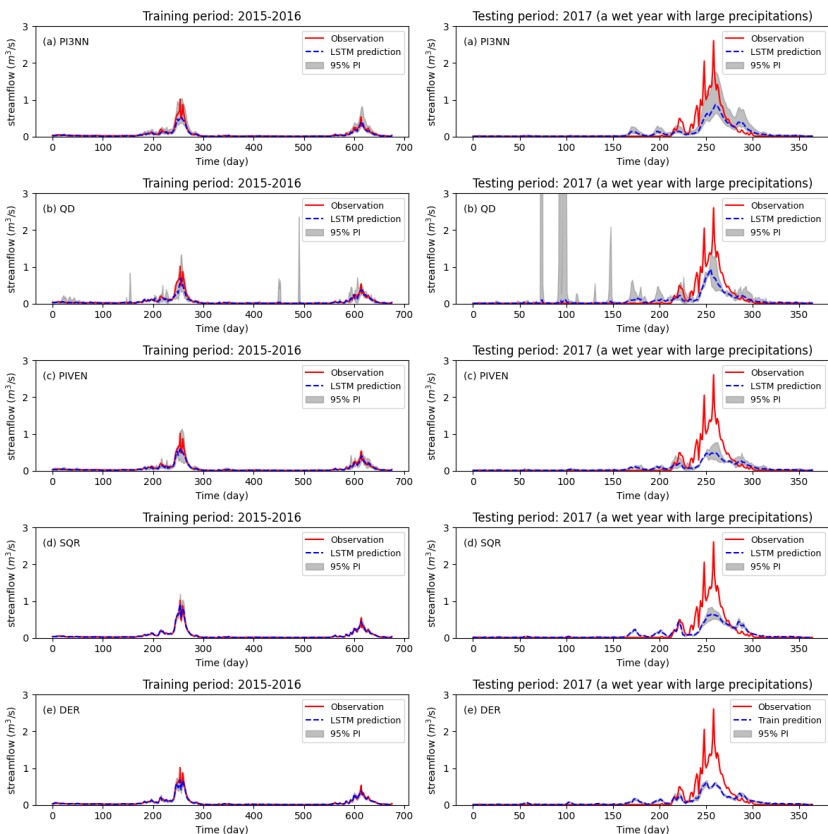

**Figure 3:** Streamflow prediction using LSTM network and the calculated 95% PIs from our PI3NN method and the baselines. The testing period (2017) is a wet year (extreme event) having dramatically different streamflow patterns than the training period. PI3NN identifies this OOD patterns by giving the prediction a large PI while other methods fail to identify the OOD patterns by giving the testing data similarly narrow PIs as the training.

