# OpenReview forum: "PI3NN: Out-of-distribution-aware Prediction Intervals from Three Neural Networks"
_ICLR.cc/2022/Conference — ICLR 2022 Poster_

### Official Review · Reviewer_AYgp · 2021-11-02

**Correctness:** 3
**Technical Novelty And Significance:** 3
**Empirical Novelty And Significance:** 3
**Recommendation:** 6
**Confidence:** 4

**Main Review:**

# Strengths

By empirical experiments, PI3NN showed the advantages over existing SOTA approaches. And the theoretical analysis and proven properties of PI3NN are provided.

# Weaknesses

Here are some of my comments/questions. I would appreciate it if the author(s) could give a response. If I am wrong, please  correct me.

- Although the design of Eq. (2), as stated by the author(s), can guarantee uniqueness, does such a design have practical validity, or is it just a theoretical trick?

- How to determine $c$ in Section 3.3 for empirical experiments? Is there any basis for selection? How do different $c$'s affect the results?


In addition, this paper is generally well written, but some places (some issues) in this paper should be further clarified/fixed.


- The format of many references is inconsistent, for example, Refs.[3] and [4]

- Some references lack journal information, for instance, Ref. [20], please provide the journal information such as

   - H. Wang and D.-Y. Yeung, “A survey on Bayesian deep learning,”  ACM Computing Surveys (CSUR), 53(5), pp.1-37, 2021

- In this paper, especially in Paragraph 2, if the abbreviations are not widely used and standard, please use the full name and abbreviations when first referring to these methods, just as Paragraph 1 of this paper adds the abbreviation "NNs" to the name of neural networks.

- In Paragraph 2, the introduction of related work is grammatically confused and inconsistent.


**Summary Of The Paper:**

Quantifying the uncertainty of neural networks is a challenge. In this paper, the author(s) proposed a method for calculating prediction intervals (PIs) based on three neural networks (PI3NN).  And the author(s) claimed that this paper addressed the limitations of existing PI methods including crossing issues, the requirement of extra hyperparameters, and lack of out-of-distribution (OOD) identification capability.  In addition, as demonstrated by the author(s) that empirical experiments showed superior performance of the proposed method over state-of-the-art (SOTA) approaches.

**Summary Of The Review:**

As claimed by the author(s), this paper addressed some of the limitations of the existing approaches. Empirical experiments showed the superiority of PI3NN, and theoretical proven properties of PI3NN are provided. So I think this is an interesting paper.

---

> ### Author Response · Authors · 2021-11-18
> **Response to comments from Reviewer AYgp**
>
> 1. The design of Eq. (2) has practical validity. Based on the design of Eq. (2), we can exactly and uniquely calculate the prediction interval that precisely encloses $\gamma$ portion (e.g., 90%) of data where 5% of data are above the upper bound and 5% of data are below the lower bound. Specifically, according to Eq. (2), we construct Eq. (10), and we then use the bisection method to calculate the optimal value of $\alpha$ and $\beta$ by finding the roots of the Eq. (10). The $\alpha$ and $\beta$ uniquely determine the prediction interval that satisfies Eq. (2).
>
>
> 2. We do not need to determine c exactly as long as c is set to a large value so that we can obtain a large output of the initial $u_{\theta}(x)$ and $l_{\xi}(x)$. As explained in the paper, a large c will cause a wider prediction interval (PI) in the OOD region than in the InD region. How much wider depends on the value of c, but we do not need to exactly quantify the magnitude of the PI in OOD region as long as it is wider than the PI in the InD region for the OOD identification. In fact, we cannot exactly quantify the uncertainty of the OOD data as we do not have a label for these data. To summarize, (1) the value of c affects the wideness of PI in OOD region but not in InD region; the larger the value of c, the wider the PI in OOD region. (2) the exact value of c would not affect the OOD identification as long as it is given a large value, for example, c=10 should be fine for most problems.
>
>
> 3.1 The format of references is now consistent.
>
> 3.2 The reference now has complete information.
>
> 3.3 The abbreviation has been spelled out.
>
> 3.4 The gramma has been corrected and made consistent.

---

### Official Review · Reviewer_QDYK · 2021-11-02

**Correctness:** 4
**Technical Novelty And Significance:** 2
**Empirical Novelty And Significance:** 2
**Recommendation:** 5
**Confidence:** 3

**Main Review:**

The targeted goal is interesting and meaningful and the authors achieved this by calculating prediction intervals for confidence levels.  The proposed methodology is well-described and introduced experimental settings are properly designed with multiple public datasets. The experimental results seem to be reasonable.

The paper addresses the import issue of uncertainty quantification, which can hugely affect real-world applications. The introduced experimental settings seem to be reasonable to show the effectiveness of its method in calculating prediction intervals and the OOD identification capability.  Although interesting, It’s a bit hard to say the introduced technique is quite new.

I believe it will be better if the authors perform the ablation studies with different parameters and more datasets with different data structures (such as images or time-series data) to generalize their findings.

**Summary Of The Paper:**

The authors propose neural networks-based method which calculate prediction intervals for uncertainty quantification. The authors validate their method with its baselines including QD, PIVEN, SQR, and DER on nine UCI datasets and  flight delay dataset. The authors suggest that their method can calculate prediction intervals without retraining neural networks and the estimated prediction intervals can avoid the crossing issue.


**Summary Of The Review:**

Although the targeted problem is interesting and the proposed method is mathematically correct with reasonable experimental results, I think more analysis is needed to make their method more interesting. It's a reasonable draft of a promising line of work.

*Response

Although this paper has a good motivation and the authors improve the quality of experiments, it's still a bit hard to say the proposed method is novel enough as it is a combination of pre-existing methods.

---

> ### Author Response · Authors · 2021-11-18
> **Response to comments from Reviewer QDYK**
>
> We use detail theoretical proof and analysis to explain that why our PI3NN method is a good uncertainty estimator and why it can identify OOD samples. Essentially, we perform a theoretical ablation study in these analyses. Additionally, we perform the ablation studies in numerical experiments. For example, we test multiple scenarios with different hyper-parameter configurations and demonstrate the accuracy and reliability of our method. Furthermore, we demonstrate that PI3NN has the OOD identification capability because of the bias initialization strategy. We use both figures and tables show the difference in the prediction results with this OOD identification capability turned on and turned off.
>
>
> In the revision, we added one more streamflow time-series data from field measurements and used Long Short-Term Memory (LSTM) network for prediction. Consistent with other datasets, PI3NN outperformed the baselines in terms of accuracy and OOD identification. Please see our response at the top for details.

---

> ### Author Response · Authors · 2021-11-28
> **Confusion about the reviewer's feedback**
>
> We thank the reviewer for taking a look at the additional experiments and our response. However, we are very confused about the reviewer's overall evaluation: "... it's still a bit hard to say the proposed method is novel enough as it is a combination of pre-existing methods". Our question is: "from the reviewer's perspective, which component of the proposed PI3NN method uses which component of the which SOTA PI method?
>
> We would like to emphasize the major differences between our method and the SOTA PI methods: (1) the SOTA PI methods use customized and complicated loss functions and our method just use very simple MSE loss, which simplifies the training process. (2) Our strategy for OOD detection (even though it's simple) is completely new. This strategy has shown significant improvement of OOD detection when using PI methods. (3) The root-finding method is also for the first time used to determine uncertainty bounds.
>
> We would really appreciate if the reviewer can provide more specific information/evidences that led to the reviewer's overall evaluation.

---

### Official Review · Reviewer_kjPV · 2021-11-02

**Correctness:** 4
**Technical Novelty And Significance:** 3
**Empirical Novelty And Significance:** 2
**Recommendation:** 6
**Confidence:** 4

**Main Review:**

**Strong Points**

- Producing confidence estimates along with predictions is clearly an important problem that deep learning methods struggle with.
- The method is more efficient than existing approaches that require retraining for different confidence levels.
- The mathematical details are easy to follow and complement the paper.
- PI3NN outperforms the baselines considered in their experiments.


**Weak Points**

- The method makes the assumption that noise is homoscedastic, which is often not the case in deep learning applications—although competing methods make this assumption also.
- The baselines compared against could be more extensive. Also see the question about the use of Equation (12) below.


**Questions**

- Did you try training with L1 loss to predict the median rather than the method outlined in Step 2 from Section 3.2?
- u_\theta and l_\xi are written as functions of x in Section 3.2. However, aren’t you assuming the targets are independent of x? This is even mentioned in the proof of Lemma 1.
- For the other methods compared against in Table 3, are they intended for OOD detection? It seems like you are comparing their performance based on the metric you define in Equation (12), but I’m unclear on whether this is reasonable. I’m curious if there are other methods for OOD detection that do not use PI that could be compared against.


**Additional Feedback**

- The assumption that the noise is homoscedastic should be discussed in the paper’s introduction.


**Summary Of The Paper:**

**Summary**

The paper proposes a new method, PI3NN, for predicting confidence intervals with neural networks. The method trains three neural networks with different loss functions, which can then be combined without retraining to give intervals for with different confidence levels. With an additional adjustment to initialization and training, the authors propose a simple way to perform OOD detection with the predicted intervals.

**Summary Of The Review:**

**Recommendation**

If my questions and concerns below are addressed, I think the paper is interesting enough for acceptance.

---

> ### Author Response · Authors · 2021-11-18
> **Response to comments from Reviewer kjPV**
>
> We appreciate the reviewer's comments. Below is our response to the reviewer's concerns.
>
> 1. The assumption of the homoscedastic noise is commonly used in regression problems. We used this assumption to simplify the theoretical proof of our method. But in practice, our PI3NN method can be generally applied to problems with different forms of noise. As shown in the paper, we applied PI3NN to nine UCI benchmark datasets and the flight delay dataset. And in this revision, we also added a streamflow time series data. Those data are all from actual observations which do not necessary have homoscedastic noise. In all these datasets, PI3NN showed superior performance to existing state-of-the-art approaches.
>
>
> 2. In the revision, we add one more streamflow time-series data from field measurements and use Long Short-Term Memory (LSTM) network for prediction. Consistent with other datasets, PI3NN outperforms the baselines in terms of accuracy and OOD identification. Please see our response at the top for details.
>
>
> 3. We trained all the three neural networks (NNs) using MSE loss (i.e., L2 loss). After we train the first NN, we obtained the mean prediction. We then followed Step 2, using a root-finding method to calculate the shift v and then adding the v to mean prediction to obtain the median approximation. In this Step 2, it did not involve NN training.
>
>
> 4. It is true in Section 3.1. that the expectation in Equation (5) is theoretically independent of $x$, because the random noise $\epsilon$ is independent of $x$. It is also true in Step 3 of Section 3.2 that the  $u_\theta$ depends on $x$ because we do not know the exact value of $\epsilon$ to train $u_\theta$. Instead, we use dataset $D_{upper}$ to train $u_\theta$ and learn/approximate the expectation where the samples in D_upper are calculated by $y-f_w(x)-v$ which are dependent on $x$, because $y-f_w(x) \neq y-f(x)$. These notations are reasonable in both theoretical and practical settings. And the theoretical justification and the algorithms are nicely connected. In Lemma 1, we prove that there exists a unique $\alpha$ to obtain the upper bound U, and in practice, we use the bisection method to uniquely determine the optimal value of $\alpha$.
>
>
> 5. PI3NN is a UQ method for NN models. An accurate UQ method should be able to capture a specified portion of data (i.e., the PICP value is close to the desired confidence level) and can also identify the OOD samples (by giving them a large uncertainty bound). Table 3 compares PI3NN with the state-of-the-art (SOTA) UQ methods and focuses on their OOD identification capability. Among the four SOTA methods, QD, PIVEN, and SQR are prediction interval (PI) methods, and DER is evidential regression approach with Gaussian assumption. The authors of DER paper claimed that DER had OOD identification capability and used probability density function plots (like our Figure 2) to demonstrate this capability. Therefore, we not only compare our method against the PI approaches, but also compare it against the DER method that has OOD detection but does not use PIs. For PI methods, Equation (12) is a reasonable metric because its calculation depends on the PI width.
>
>
> 6. The discussion of the homoscedastic noise assumption has been added in the revised paper.

---

> > ### Comment · Reviewer_kjPV · 2021-11-26
> > **Response**
> >
> > I am grateful to the authors for their response to my question. The comparison between different methods of OOD detection seems reasonable. I feel the paper would be of interest to attendees of ICLR and that the paper makes a nice contribution.
> >
> > The reason I asked about using L1 loss to train the NN rather than L2 is that the minimizer is the media in this case, which might perform better than Step 2. The discussion around the dependence on x in Section 3.2 could still be made clearer. Given this I will keep my score as is.

---

### Official Review · Reviewer_TzHb · 2021-11-02

**Correctness:** 3
**Technical Novelty And Significance:** 3
**Empirical Novelty And Significance:** 2
**Recommendation:** 5
**Confidence:** 3

**Main Review:**

+ves:

- The method seems to achieve its motivating goals mentioned above.

- The paper has considered OOD detection and proposes an idea to help with that along some experiments.

- The experiments show that the performance of the method is comparable or slightly better than competitive baselines and less sensitive to some hyperparameters.

Concerns:
- One of my main concerns is that there seems to be a disconnect between the theoretical justification and the algorithm. For example, based on the assumptions in Section 3.1, the expectation in (5) is independent of x. Going to step 3 in 3.2, the term is considered to be dependent on x. (9) does not seem to agree with (5).

- Several setups of different methods have similar PICP values. It is not clear why PI width cannot be compared among them.

- The experiments are mostly on toy or simple data and architectures. There is no experiment on other types of data like images.

- The idea to help OOD detection is limited to single target ReLU networks.

- While one motivation is to train the neural networks once, there is no experiment to show results for different confidence intervals for fixed trained networks.

- The notation is unfortunately confusing at times. For example, given or at a specific x is inconsistently dropped or included.

- One disadvantage is that the method is restricted to one dimensional output while some of the other baselines are more versatile.

- There is no discussion about the scalability and run time of the method.


Minor:

- There seems to be a typo in Theorem 1 (> and < flipped)


**Summary Of The Paper:**

The paper proposes a new prediction interval method which is called prediction interval based on three neural networks (PI3NN). The motivation is to have a method that neither requires retraining neural networks for different confidence levels nor involves highly customized loss functions. In PI3NN three neural networks are trained once and a linear combination of their outputs is used for predicting intervals. Also, a variation of the method is proposed for OOD detection.

**Summary Of The Review:**

The paper is well motivated, but the theoretical analysis does not appear to be solid enough for the proposed method and the experiments are limited.

---

> ### Author Response · Authors · 2021-11-18
> **Response to comments from Reviewer TzHb**
>
> We appreciate the reviewer's comments. Below is our response to the reviewer's concerns.
>
> 1. It is true in Section 3.1. that the expectation in (5) is theoretically independent of $x$, because the random noise $\epsilon$ is independent of $x$. It is also true in Step 3 in Section 3.2 that the  $u_\theta$ depends on $x$ because we do not know the exact value of $\epsilon$ to train $u_\theta$. Instead, we used dataset $D_{upper}$ to train $u_\theta$ and learn/approximate the expectation where the samples in $D_{upper}$ are calculated by $y-f_w(x)-v$ which are dependent on $x$ in practice, because $y-f_w(x)$ depends on $x$ (because $f_w(x)$ is not equal to $f(x)$). These notations are reasonable in both theoretical and practical settings. And the theoretical justification and the algorithms are nicely connected. In Lemma 1, we proved that there exists a unique $\alpha$ to obtain the upper bound $U$, and in practice, we used bisection method to uniquely determine the optimal value of $\alpha$.
>
>
> 2. As we explained in the paper, it is meaningful to compare PI widths only when the methods have the same PICPs. In the newly added Table 4 of Appendix D, we listed the mean PI width (MPIW) for the methods having similar PICPs. Essentially, Table 4 listed the MPIWs for the best PICPs highlighted in Table 1 of the paper. Table 4 indicates that for the similar PICPs, our PI3NN methods gave the smallest MPIWs for most of the datasets (7 out of 9). Please note that PI3NN obtained small MPIWs using the standard MSE loss, while other PI methods such as QD and PIVEN specifically minimized MPIWs in their loss function. This customized loss although sometimes gives better MPIWs after careful hyper-parameter tuning, it also results in unreliable prediction performance sensitive to the hyper-parameter configuration (as discussed in Table 1 of the paper).
>
> 3. We added a new time-series data, see our response at the top.
>
>
> 4. We clarify that the OOD detection is NOT limited to single target ReLU networks. In the paper, we used the ReLU network as an example (due to its wide application). For other type of activation functions, the following situation is generally true. The bias of the output layer acts as a shifting parameter. Initializing the bias to a large value will significantly increase the initial output values of the network. During training, the loss will encourage the decrease of upper-bound and lower-bound networks only for the InD samples, not for the OOD samples. Then after training, the prediction interval (PI) will be wider in the OOD region than in the InD region. Additionally, due to continuity of the network, the PI width will increase with the distance between $x$ and $D_{train}$ showing a decreasing confidence.
>
>
> 5. In Section 3.2, we theoretically proved that PI3NN can generate PIs for multiple confidence levels ($\gamma$) without re-training the neural networks (NNs) and the produced PIs have non-crossing property. In Figure 1, we used an experiment to show results for different confidence levels for fixed trained networks. In Section 3.4, we clearly explained and demonstrated that after training the three NNs, for a given series of confidence levels, we used the root-finding technique to calculate the corresponding $\alpha$ and $\beta$ values in Step 4 and obtained the associated e.g., 90%, 95%, and 99% PIs as shown in Figure 1.
>
>
> 6. In this revision, the notation has been carefully checked to be consistent.
>
>
> 7. PI3NN can be applied to multiple outputs, and the application is straightforward. In training the three NNs, we can construct the NNs to learn the relationship between the inputs and multiple outputs. Then after NN training, for each output, we calculate the corresponding $\alpha$ and $\beta$ to determine its PI. Currently, the other PI methods such as QD, PIVEN, and SQR only applied to a single output, and it is not straightforward to apply them to multiple outputs. For example, for a single output, QD and PIVEN calculate its PI by specifically minimizing this output's PI width in the loss, i.e., their loss function was customized to calculate the PI of the one-dimensional output. Then, for multiple outputs, QD and PIVEN need to calculate the PI of a single output one by one, each calculation involves one NN training. On the other hand, our PI3NN method only needs one-time NN training to calculate the PIs for multiple outputs.
>
>
> 8. To calculate PIs for one or multiple outputs, PI3NN only needs to train 3 NNs with the standard MSE loss. This requires a standard time of a MSE-network training. Then after training, we use the bisection method to calculate PIs for any number of outputs, and for each output we calculate PIs for any given number of confidence levels without retraining the NNs. The computing time of bisection root-finding is very fast (usually in seconds), as proven in the numerical mathematics textbook (Quarteroni et al., 2007)
>
>
> 9. The typo in Theorem 1 has been corrected.

---

> > ### Comment · Reviewer_TzHb · 2021-11-27
> > **After author response**
> >
> > I would like to thank the authors for their responses. Further experiments, results, and clarifications have improved the paper and addressed some of my concerns. I have increased the score accordingly.
> >
> > But unfortunately, I think my concern regarding the disconnect between the theoretical justification and the algorithm is not adequately addressed. I think the authors can discuss whether the theoretical justification holds if the expectation in (5) is dependent on x or the homoscedasticity assumption is violated. Also, they can further discuss why currently the estimate of the expectation is dependent on x and clarify whether under such a circumstance the key theoretical justification for having alpha independent of x holds. The authors seem to suggest in their response that an x-dependent estimate for the expectation is due to uncertainty/error of median estimation, but this is also not clearly and rigorously discussed in the current presentation and connected to Section 3.1 and Lemma 1.

---

> > > ### Author Response · Authors · 2021-11-28
> > > **Additional response**
> > >
> > > We really appreciate the reviewer's feedback. Below is our additional responses to the reviewer's comments.
> > >
> > > 1. When the homoscedasticity assumption is violated, we agree that Lemma 1 will not hold anymore. Even though our PI3NN algorithm can still provide a PI prediction, but the prediction will not converge to the true PI for every $x$ from theoretical perspective. However, to our knowledge, none of the SOTA PI methods provide rigorous convergence proof of their PI predictions to the ground-truth PIs in the heteroscedastic context. Thus, the performance of PI methods for heteroscedastic problems are usually compared using numerical experiments. As shown in Section 4, the UCI benchmark datasets do not satisfy the homoscedastic assumption, and our PI3NN method outperforms the SOTA PI methods, which experimentally demonstrates the advantages of our method.
> > >
> > > 2. When the homoscedasticity assumption holds, the dependence of $u_\theta$ and $l_\xi$ on $x$ is due to the error of median estimation. To explain it clearly, we will change the first sentence of Step 3 in Section 3.2 to
> > >
> > > "Train $u_\theta(x)$ to learn $\mathbb{E}[(y-f_\omega(x) - \nu) \mathbf{1}_{y-f_\omega(x) - \nu >0}]$
> > >
> > >  and train $l_\xi(x)$ to learn $\mathbb{E}[(f_\omega(x) + \nu-y) {1}_{f_\omega (x) + \nu - y >0}]$"
> > >
> > > In this way, it is easy to see that $y-f_\omega(x) - \nu$ depends on $x$ because $f_\omega(x) - f(x)$ depends on $x$.
> > >
> > > 3. We agree that theoretical analysis of PI methods for the heteroscedastic setting is important to study, and we will add additional discussion in Conclusion section to clearly show that having no theoretical proof in the heteroscedastic setting is a limitation of our method. However, since none of the SOTA PI methods provides rigorous convergence proof in the heteroscedastic setting, we argue that the acceptance of this paper should not be judged by whether our theoretical justification is applicable to the heteroscedastic setting.
> > >
> > > We would be happy to answer any additional comments.

---

### Author Response · Authors · 2021-11-19
**Response to comment on adding extra dataset and numerical experiments**

In the revision, we added one more streamflow time-series data from field measurements in East River Watershed of Colorado, United States, and compared PI3NN prediction performance against the baselines. The dataset contains three years of daily data of precipitation, maximum temperature, minimum temperature, and streamflow at the catchment outlet. We use the Long Short-Term Memory (LSTM) network to learn the relationship between the three meteorological forcing (i.e., precipitation, maximum temperature and minimum temperature) and the streamflow. The training data are from the first two years (2015-2016) and the third year (2017) of data form the testing set. Note that 2017 is a wet year with a relatively large amount of precipitation, which causes the streamflow patterns in 2017 dramatically different from those in 2015-2016. The newly added Figure 3 in Section 4.4 indicates that PI3NN can accurately identify the OOD data (i.e., the extreme event in 2017) by giving their predictions a large PI, while other methods fail to identify the OOD patterns by giving the testing data similarly narrow PIs as the training data. This example demonstrates that our PI3NN method can show a consistent superior performance for a different dataset (time-series data) and a different network architecture (LSTM network).

---

### Decision · Program_Chairs · 2022-01-20

**Decision:**

Accept (Poster)

**Comment:**

The paper proposes a novel method, PI3NN, for estimating prediction intervals (PIs) for quantifying the uncertainty of neural network predictions. The method is based on independently training three neural networks with different loss functions which are then combined via a linear combination where the coefficients for a given confidence level can be found by the root-finding algorithm. A specific initialization scheme allows to employ the method to OOD detection.

Reviewers agreed on the importance of the problem of producing reliable confidence estimates.  The proposed method addressed some of the limitations of the existing approaches, and reviewers valued that a theoretical as well as an empirical analysis is provided.

On of the main criticisms was that the theoretical derivation of the method is based on the assumption of the noise being homoscedastic. This however is a common issue with other methods in this area, which are nevertheless all applied (and seem to work) on heteroscedastic data as well and are outperformed by the proposed method. Another main point that was criticized was that the empirical analysis was limited. In turn the authors added another experiments on another dataset and with another network architecture (a LSTM) to their analysis. Moreover, the authors adequately addressed a lot of the concerns and questions of the reviewers in their answers and the revised manuscript.  The final mean scores are exactly borderline (5.5) but with a higher confidence of reviewers voting for acceptance.  Based on the listed points, the paper should be accepted.

I would encourage to  improve the discussion around the dependence on x in Section 3.2, which could still be made clearer, in the final version of the manuscript, and to add the discussion about the limitations of the theoretical analyses (i.e. the applicability  only to the homoscedastic settings) to the conclusion.